# Dual-Energy CT in Acute Stroke: Could Non-Contrast CT Be Replaced by Virtual Non-Contrast CT? A Feasibility Study

**DOI:** 10.3390/jcm13133647

**Published:** 2024-06-21

**Authors:** Guillaume Herpe, Alexandra Platon, Pierre-Alexandre Poletti, Karl O. Lövblad, Paolo Machi, Minerva Becker, Michel Muster, Thomas Perneger, Rémy Guillevin

**Affiliations:** 1Emergency Radiology Unit, Division of Radiology, University Hospital of Geneva, 4 Rue Gabrielle-Perret-Gentil, 1205 Geneva, Switzerland; alexandra.platon@hcuge.ch; 2DACTIM-MIS Lab, I3M, Poitiers University, 86021 Poitiers, France; remy.guillevin@chu-poitiers.fr; 3Division of Radiology, University Hospital of Geneva, 4 Rue Gabrielle-Perret-Gentil, 1205 Geneva, Switzerland; pierre-alexandre.poletti@hcuge.ch (P.-A.P.); minerva.becker@hcuge.ch (M.B.); 4Division of Neuroradiology, University Hospital of Geneva, 4 Rue Gabrielle-Perret-Gentil, 1205 Geneva, Switzerland; karl-olof.lovblad@hcuge.ch (K.O.L.); paolo.machi@hcuge.ch (P.M.); michel.muster@hcuge.ch (M.M.); 5Division of Clinical Epidemiology, University Hospital of Geneva, 4 Rue Gabrielle-Perret-Gentil, 1205 Geneva, Switzerland; thomas.perneger@hcuge.ch

**Keywords:** dual-energy CT, acute ischemic stroke, virtual non-contrast CT

## Abstract

**Purpose:** We aimed to evaluate whether virtual non-contrast cerebral computed tomography (VNCCT) reconstructed from intravenous contrast-enhanced dual-energy CT (iv-DECT) could replace non-contrast CT (NCCT) in patients with suspected acute cerebral ischemia. **Method:** This retrospective study included all consecutive patients in whom NCCT followed by iv-DECT were performed for suspected acute ischemia in our emergency department over a 1-month period. The Alberta Stroke Program Early CT Score (ASPECTS) was used to determine signs of acute ischemia in the anterior and posterior circulation, the presence of hemorrhage, and alternative findings, which were randomly evaluated via the consensus reading of NCCT and VNCCT by two readers blinded to the final diagnosis. An intraclass correlation between VNCCT and NCCT was calculated for the ASPECTS values. Both techniques were evaluated for their ability to detect ischemic lesions (ASPECTS <10) when compared with the final discharge diagnosis (reference standard). **Results:** Overall, 148 patients (80 men, mean age 64 years) were included, of whom 46 (30%) presented with acute ischemia, 6 (4%) presented with intracerebral hemorrhage, 11 (7%) had an alternative diagnosis, and 85 (59%) had no pathological findings. The intraclass correlation coefficients of the two modalities were 0.97 (0.96–0.98) for the anterior circulation and 0.77 (0.69–0.83) for the posterior circulation. The VNCCT’s sensitivity for detecting acute ischemia was higher (41%, 19/46) than that of NCCT (33%, 15/46). Specificity was similar between the two techniques, at 94% (97/103) and 98% (101/103), respectively. **Conclusions:** Our results show that VNCCT achieved a similar diagnostic performance as NCCT and could, thus, replace NCCT in assessing patients with suspected acute cerebral ischemia.

## 1. Introduction

Brain stroke is the second cause of mortality across the world and the third cause of disability [1]. The constant ongoing development of stroke units and the centralization of acute stroke services have increased the number of patients undergoing imaging for suspected stroke [2].

Both magnetic resonance imaging (MRI) and computed tomography (CT) are effective tools in acute stroke management. However, given that CT is more easily available and less affected by patient motion than MRI, this technique remains the most frequently used imaging modality for assessing patients with suspected stroke [3].

CT protocols for stroke usually include non-contrast cerebral CT (NCCT), followed by an intravenous (IV) injection of contrast media for CT perfusion, CT angiography, and delayed post-contrast series [4].

Recent advances in dual-energy CT (DECT) technology, based on the attenuation patterns of different body structures at various tube voltages, have opened up new perspectives in the diagnostic management of patients with acute stroke. Indeed, virtual non-contrast CT (VNCCT) images can be obtained by subtracting iodine from IV contrast-enhanced DECT (iv-DECT) images [5,6]. Even though the quality of VNCCT images has greatly improved in recent CT scanners, no prior series, to the best of our knowledge, have specifically compared NCCT images with VNCCT images reconstructed from iv-DECT series in patients admitted with a suspicion of acute stroke [5].

Non-contrast computed tomography (NCCT) is a widely used imaging modality for the initial evaluation of suspected acute cerebral ischemia due to its rapid availability and ability to exclude hemorrhage [7]. However, NCCT has limitations in sensitivity for detecting early ischemic changes and small infarcts. Dual-energy computed tomography (DECT) is an advanced imaging technique that differentiates materials based on their energy-dependent absorption characteristics. By acquiring images at two different energy levels, DECT enables the reconstruction of virtual non-contrast CT (VNCCT) images from contrast-enhanced scans. This capability offers the potential to replace conventional NCCT with VNCCT, potentially enhancing diagnostic accuracy and workflow efficiency in acute stroke settings. This study aims to evaluate whether VNCCT, reconstructed from intravenous contrast-enhanced dual-energy CT (iv-DECT), can effectively replace NCCT in patients with suspected acute cerebral ischemia. We hypothesized that VNCCT would demonstrate comparable diagnostic performance to NCCT in detecting ischemic lesions, as measured by the Alberta Stroke Program Early CT Score (ASPECTS) [8] and the presence of hemorrhage and alternative findings. While previous studies have explored the utility of DECT in various clinical settings, the application of VNCCT, specifically for acute stroke diagnosis, remains under-researched. 

The goal of our current study was to evaluate whether VNCCT could replace true NCCT in the diagnostic work-up of patients admitted for suspected acute stroke.

## 2. Materials and Methods

This retrospective study was approved by the hospital’s institutional review board (IRB 2018-00476, approved on 9 August 2018) without any patients’ informed consent being required.

Our analysis included cerebral CT images from all consecutive patients over 18 years old who underwent brain CT examinations for suspected acute cerebral stroke at the Geneva University Hospital emergency radiology unit. These data were collected over a one-month period from 1 November 2022, at 00:01 AM to 30 November 2022, at 11:59 PM.

### 2.1. CT Protocol

In our institution, all patients with suspected acute stroke underwent a standardized CT examination comprising cerebral NCCT, CT angiography of the supra-aortic vessels, and delayed cerebral CT using a dual-energy technique.

The examinations were performed on a dual-source Somatom Force CT machine (Siemens Healthineers, Erlangen, Germany). NCCT was obtained from the vertex to the foramen magnum at an acquisition thickness of 3–3 mm, pitch of 0.55, rotation time of 1 s, 120 kV, automated mAs modulation, and 3–1 mm reconstruction. CT angiography was obtained from the left atrium to the vertex after injecting 70 cc of iodine contrast material (Iohexol, 350 mg I/mL, Accupaque^®^, GE Healthcare, Chicago, IL, USA) at a rate of 4 mL/s, with the region of interest in the ascending aorta, and automatic triggering at 150 HU. The examination was performed using an acquisition thickness of 0.75–0.5 mm, pitch of 0.7, rotation time of 0.25 s, 90–150 kV, automated mAs modulation, and 0.75–0.5 mm reconstruction.

Delayed post-contrast cerebral DECT was obtained from the vertex to the foramen magnum at 180 s following the contrast medium injection, using 80–150 kV, 3–3 mm acquisition thickness, 0.55 pitch, 1 s rotation time, automated mAs modulation, and 3–3 mm reconstruction.

### 2.2. VNCCT Reconstruction

At the end of the examination, VNCCT images were obtained by automatically post-processing the iv-DECT images on a dedicated workstation (Syngo.via VB20A_HF05, Siemens Healthineers, Erlagen, Germany), using an intra-cranial hemorrhage algorithm at an iodine level of 3.01 and a 3/1 mm slice thickness.

### 2.3. Image Analysis

VNCCT and NCCT were retrospectively analyzed by means of consensus reading by two board-certified radiologists from the emergency radiology unit who had 10 and 5 years of experience in reading brain CTs, respectively. Consensus reading was employed to reproduce the reading routine in our emergency unit. Readers were blinded to the patient’s clinical data and to their final diagnosis. All cases were anonymized, and the reading was performed in random order. The CT angiography and iv-DECT images were not available to the readers. CT images were interpreted using an OsiriX workstation (Pixmeo, Bernex, Switzerland).

A dedicated stroke window [9] was applied in all cases, which could be adjusted by the readers as necessary.

The following CT signs were recorded:Parenchymal hypodensities are characteristic of an acute ischemic event in the anterior and posterior circulation. These were defined according to the Alberta Stroke Program Early CT Score (ASPECTS) [3,10], which rates ischemia in the middle and posterior cerebral artery territories on a scale of 0 to 10, 10, indicating that no signs of ischemia are visible [5,11,12,13]. The examination was considered positive for early ischemic changes if the ASPECTS was under 10 [12].Intra-arterial thrombus is defined as a hyperdense artery in the anterior or posterior cerebral circulation;Intra- or extra-axial hemorrhages are defined as hyperattenuating areas of at least 60 HU [6].

### 2.4. Reference Standard

For each patient, the final diagnosis (considered as reference standard) was taken from the discharge summary, which included the radiological follow-up. A 120-day clinical and radiological follow-up was obtained for every patient who did not undergo control imaging to screen for alternative diagnoses.

### 2.5. Statistical Analysis

Statistical analysis was performed using SPSS software (IBM^®^ SPSS^®^ Statistics 22, IBM Corporation, Armonk, NY, USA). We obtained two-way-mixed, absolute agreement, single-measure intraclass correlation coefficients for VNCCT and NCCT with 95% confidence intervals. We examined the distribution of the scores according to the presence or absence of stroke in the corresponding anterior and posterior circulation areas.

With regard to the extent of ischemia, ASPECTS was subdivided into two categories using a cutoff score of 8. ASPECTS < 7 were frequently associated with severe ischemia and bad prognosis, while scores between 8 and 10 indicated potentially salvageable tissue [12]. The ability of VNCCT and NCCT to demonstrate ischemia was compared to the reference standard to calculate sensitivity and specificity. The statistical significance threshold was defined as a *p*-value < 0.05.

## 3. Results

Of the 148 patients, 80 (54.1%) were men. Their age ranged from 22 to 100 years (mean 64.8).

A final diagnosis of acute ischemia was made in 46 (31%) of the 148 patients, while 6 patients (4%) had intracerebral hemorrhage, 11 (7%) had an alternative diagnosis, and 85 (58%) had no pathological findings (Figure 1).

-Acute ischemia: found in 46 patients (30.5%).-Anterior circulation: found in 39 patients (85.5%); detected in 16 patients on virtual non-contrast CT (VNCCT) and 12 on non-contrast CT (NCCT).-Posterior circulation: found in 8 patients (15.5%); detected in 6 patients on VNCCT and 6 on NCCT. Note: One patient exhibited both anterior and posterior ischemic signs.-Intraparenchymal hemorrhage: found in 6 patients (4%), all detected on both VNCCT and NCCT.-Alternative diagnoses: found in 11 patients (11%); 5 detected on VNCCT and 5 on NCCT.-Normal findings: present in 85 patients (89%), confirmed by both VNCCT and NCCT.

The distribution of ASPECTS values for the anterior and posterior circulation is reported in Table 1.

Intra-arterial thrombi (Figure 2) were observed in four patients on VNCCT and in the same four patients on NCCT. Likewise, intracerebral hemorrhage was diagnosed in six patients on VNCCT and in the same six patients on NCCT (Figure 3).

Eleven patients (7%) presented various alternative diagnoses (Table 2). Five of them (45%) were detected on both VNCCT and NCCT as lesions not suggestive of ischemia, of which four were brain tumors, and one was a cerebral abscess (Figure 4).

### 3.1. Agreement between VNCCT and NCCT

#### 3.1.1. Anterior Circulation

On NCCT, the ASPECTS for the anterior circulation ranged from 4 to 10. Most patients (136, 91.9%) received the maximum score of 10, and 7 (4.7%) received low scores between 4 and 7. On VNCCT, anterior circulation ASPECTS also ranged from 4 to 10. Most patients (132, 89.2%) received the maximum score of 10, and seven (4.7%) were given low scores between 4 and 7.

In 141 patients (95.3%), the scores were identical to the two modalities. Five patients had a discrepancy of 1 point only, with four having 10 points on NCCT and 9 points on VNCCT, while one had 9 points on NCCT and 8 points on VNCCT. One patient had a difference of 2 points, with 4 points on NCCT and 6 points on VNCCT, and one exhibited a difference of three points (7 points on NCCT and 4 points on VNCCT). Thus, only two patients (1.4%) displayed a discrepancy greater than 1 point, and neither had discrepancies between high and low scores (8–10 vs. 0–7).

The intraclass correlation coefficient of the two modalities was 0.973 (95% confidence interval: 0.963 to 0.981).

#### 3.1.2. Posterior Circulation

On NCCT, posterior circulation ASPECTS values ranged from 6 to 10. Most patients (142, 95.9%) received the maximum score of 10, two (1.4%) received low scores of 6, and none received a 7. On VNCCT, the posterior circulation scores also ranged from 6 to 10. Most patients (139, 93.9%) received a maximum score of 10, and three (2.0%) received scores of 6.

For 141 patients (95.3%), ASPECTS values were identical for the two modalities, while in three patients, the discrepancy was 1 point only (10 on NCCT and 9 on VNCCT). Two patients had a difference of two points (10 on NCCT and 8 on VNCCT, and 8 and 6), one had a difference of 3 points (9 on NCCT and 6 on VNCCT), and one had a difference of 4 points (6 on NCCT and 10 on VNCCT). Thus, only four patients (2.7%) exhibited a discrepancy greater than 1 point, and for one patient (0.7%), the discrepancy occurred between high and low scores (8–10 vs. 0–7).

The intraclass correlation coefficient of the two modalities was 0.777 (95% confidence interval: 0.691 to 0.839).

### 3.2. Performance of VNCCT and NCCT to Detect Acute Ischemia

VNCCT was reported as a positive for ischemia in 24 (16%) of 148 of the patients, 5 of whom were considered to have false positives. VNCCT was reported as negative in 124 patients, 27 of whom were considered to have false negatives. The VNCCT sensitivity was 41% (19/46), and its specificity was 94% (97/103). NCCT was considered positive for ischemia in 17 (11%) of the 148 patients, 2 of whom were considered to have false positives. NCCT was reported as negative in 131 patients, 31 of whom were considered to have false negatives. The NCCT sensitivity was 33% (15/46), and its specificity was 98% (101/103).

## 4. Discussion

Our study sought to evaluate whether VNCCT could replace NCCT for assessing patients admitted with suspected acute stroke. In this patient population, our results revealed an excellent correlation (intraclass correlation coefficients of 0.97 for the anterior circulation and 0.77 for the posterior circulation) for determining ASPECTS between VNCCT and NCCT [14]. This strong agreement suggests that both techniques are equivalent for this task. The slightly lower correlation coefficient in the posterior circulation can be attributed to the presence of multiple artifacts caused by the area’s anatomy, which renders its analysis more challenging.

Additionally, we evaluated the diagnostic performance of both VNCCT and NCCT in detecting ischemia. VNCCT demonstrated a sensitivity of 41% (19 out of 46 patients) compared to 33% (15 out of 46 patients) for NCCT, indicating a superior ability to detect ischemic lesions. Specificity was similar between the two techniques, with VNCCT showing a specificity of 94% (97 out of 103 patients) and NCCT showing a specificity of 98% (101 out of 103 patients). These findings highlight the potential of VNCCT to enhance early ischemia detection, which is critical for timely and effective stroke management [15,16].

Moreover, our study identified six cases of intracerebral hemorrhage, all detected by both VNCCT and NCCT, confirming that VNCCT is also reliable for identifying hemorrhagic events. The capability of VNCCT to provide equivalent diagnostic information without the need for separate NCCT scans suggests a more efficient workflow, reducing the need for multiple scans and thereby minimizing patient movement and scan times.

These results are consistent with previous studies, which reported sensitivities ranging from 33 to 52% and specificities ranging from 67 to 100% for depicting acute ischemia on unenhanced cerebral CT [5,9,10,14,17]. These low sensitivities for both techniques are easily explained by the fact that early ischemic signs, occurring in the first hours after stroke, can remain undetectable during CT [17].

An ASPECTS of eight has often been reported as a useful cutoff for treatment selection [18] alongside other parameters, such as clinical context and findings on enhanced CT [19,20]. However, in our 148 patients, no differences were found between the two techniques at the cutoff of eight in the anterior circulation, and only one difference was found (0.7%) in the posterior circulation.

In the current study, six brain hemorrhages were detected both on NCCT and on VNCCT. Despite this small number of hemorrhages, our observation substantiates the results of recent studies that have reported VNCCT as having both high sensitivity (92–94.5%) and specificity (97–100%) for depicting subarachnoid hemorrhages [6,21].

An alternative diagnosis was found in 7.4% of patients (11/148), which is close to the 6.4% reported in the scientific literature [22]. Our study demonstrated that VNCCT achieved the same sensitivity (45%, 5/11) and specificity (100%, 92/92) as NCCT for uncovering possible alternative diagnoses.

These findings align with previous studies on dual-energy CT (DECT) in ischemic stroke. For instance, a study comparing DECT with conventional NCCT reported a concordance correlation coefficient (CCC) of 0.44 for VNCCT, which is higher than 0.23 for conventional NCCT, highlighting VNCCT’s superior sensitivity in detecting early ischemic changes. Another study found that DECT could effectively differentiate between iodine and blood components, enhancing the detection of ischemic tissue and providing a reliable alternative to NCCT [23].

Additionally, our results are consistent with the broader literature on DECT’s prognostic value. A meta-analysis indicated that patients with contrast staining (CS) detected by DECT had a higher risk of hemorrhagic transformation (HT) and poorer functional outcomes at 90-day follow-up compared to those without CS. This underscores DECT’s utility in identifying patients at higher risk of complications, thus aiding in more informed clinical decision making [24].

Among the limitations of our study, we need to consider the lack of an immediate imaging reference standard regarding the ASPECT score. Estimating the ASPECTS later on MRI or a control CT cannot be considered as an immediate reference standard since the evolution of the ischemia is related to multiple parameters, including treatment. This limitation is, however, common given that the two techniques are usually not performed together at admission.

We must also acknowledge the retrospective analysis of the CT images. The images were read in consensus and under optimal conditions by well-trained radiologists without any stress or time constraints. Whether VNCCT would have achieved the same diagnostic performance in real-life conditions of use with less experienced radiologists remains to be confirmed. Nevertheless, consensus reading was chosen to fit what generally happens in clinical situations and seemed to be the most clinically pertinent.

Our study did not evaluate how the CT scanner technology and/or the post-processing tools might impact the image analysis. Therefore, the fact that very different results could have been achieved using different devices and software is not excluded. Further studies are thus needed to answer these important questions.

## 5. Conclusions

Our study suggests that VNCCT could safely replace NCCT for assessing patients with suspected acute stroke. The high intraclass correlation coefficients for ASPECTS in both the anterior and posterior circulation indicate a strong agreement between VNCCT and NCCT. Additionally, VNCCT demonstrated better sensitivity and comparable specificity to NCCT in detecting acute ischemia. These findings highlight the potential of VNCCT to streamline the diagnostic process, reduce patient movement, and decrease scanning times without compromising diagnostic accuracy. Further studies with larger sample sizes are warranted to confirm our preliminary positive observations and to explore the practical implementation of VNCCT in various clinical settings.

## Figures and Tables

**Figure 1 jcm-13-03647-f001:**
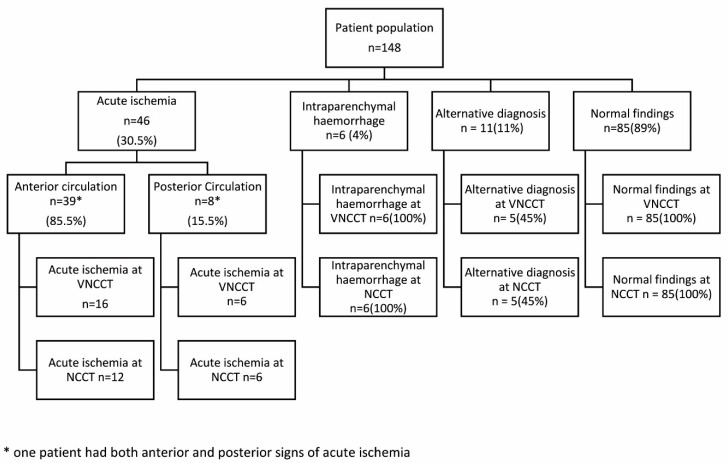
This flowchart illustrates the diagnostic outcomes of the study.

**Figure 2 jcm-13-03647-f002:**
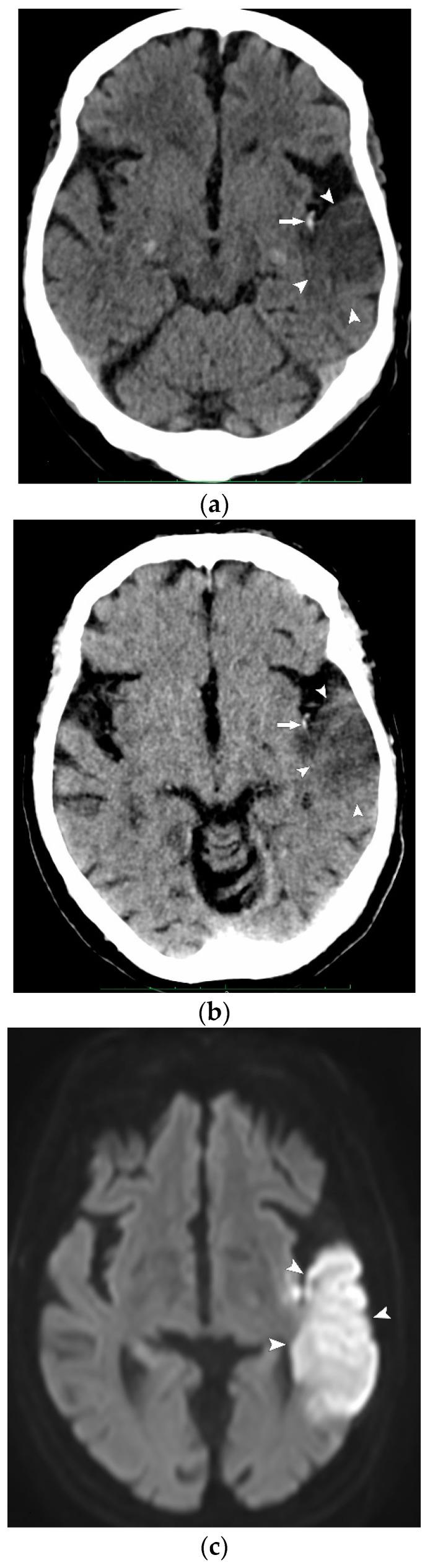
Admission CT of an 82-year-old woman presenting with acute aphasia. (**a**) NCCT and (**b**) VNCCT show a hyperdense thrombus in the M2 segment of the left middle cerebral artery (arrow), along with low-attenuation ischemic changes (arrowheads) in the left superior temporal gyrus; (**c**) follow-up diffusion-weighted MRI confirmed the occurrence of ischemia in the left temporal lobe (arrowheads).

**Figure 3 jcm-13-03647-f003:**
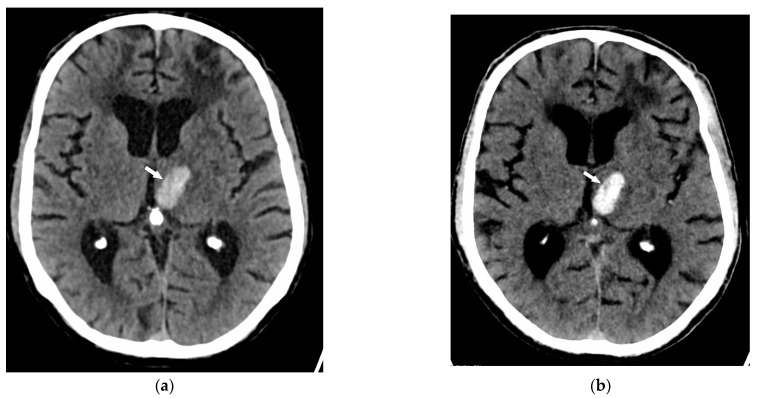
Admission CT of a 79-year-old man presenting with acute dysphasia. (**a**) NCCT and (**b**) VNCCT reveal an oval-shaped hyperdense area within the left thalamus (arrow) consistent with an acute hematoma.

**Figure 4 jcm-13-03647-f004:**
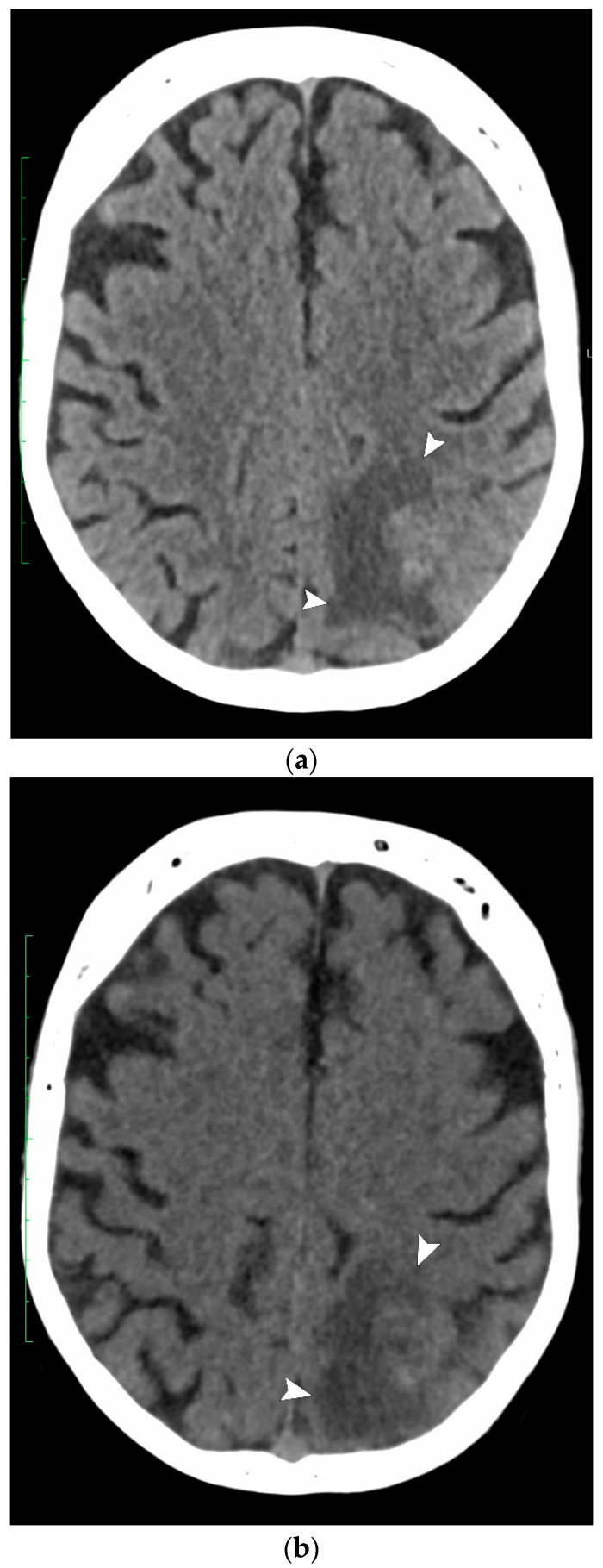
Admission CT of a 50-year-old man presenting with acute right hemiplegia. (**a**) NCCT and (**b**) VNCCT show an ill-defined hypodense area with finger-like extension into the left parietal white matter (arrowheads), which is consistent with vasogenic edema; (**c**) intravenous contrast-enhanced CT at the same level as Figure 4a,b shows an irregular, ring-enhancing lesion (arrow) surrounded by vasogenic edema (arrowheads), suggesting either an infection or a tumor. A pyogenic cerebral abscess was confirmed at follow-up.

**Table 1 jcm-13-03647-t001:** Distribution of ASPECT scores in the anterior circulation on non-contrast and virtual non-contrast computed tomography series.

ASPECTS	Anterior Circulation (*n* = 148)		Posterior Circulation (*n* = 148)	
	VNCCT	NCCT	VNCCT	NCCT
4	*n* = 3	*n* = 3	*n* = 0	*n* = 0
5	*n* = 1	*n* = 1	*n* = 0	*n* = 0
6	*n* = 2	*n* = 1	*n* = 3	*n* = 2
7	*n* = 1	*n* = 2	*n* = 0	*n* = 0
8	*n* = 3	*n* = 2	*n* = 1	*n* = 2
9	*n* = 6	*n* = 3	*n* = 4	*n* = 2
10	*n* = 132	*n* = 136	*n* = 140	*n* = 142

VNCCT, virtual non-contrast computed tomography; NCCT, non-contrast computed tomography.

**Table 2 jcm-13-03647-t002:** Detection of alternative diagnoses on virtual non-contrast and non-contrast computed tomography series.

Alternative Diagnoses *n* = 11	Diagnosis Suspected at VNCCT *n* = 5	Diagnosis Suspected at NCCT *n* = 5
Tumor *n* = 4	4	4
Idiopathic seizures *n* = 2	0	0
Posterior reversible encephalopathy syndrome *n* = 1	0	0
Multiple sclerosis *n* = 1	0	0
Meniere’s disease *n* = 1	0	0
Abscess *n* = 1	1	1
Acute psychotic disorder *n* = 1	0	0

VNCCT, virtual non-contrast computed tomography; NCCT, non-contrast computed tomography.

## Data Availability

The data presented in this study are available on request from the corresponding author.

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
