# Peer review of "Dual-Energy CT in Acute Stroke: Could Non-Contrast CT Be Replaced by Virtual Non-Contrast CT? A Feasibility Study"

_jcm, 2024, doi:10.3390/jcm13133647_

Round 1

Reviewer 1 Report

Comments and Suggestions for Authors

The article is interesting and well written.

Abstract: ok

Introduction: can you explain the rationale of routinely including a 180s post contrast acquisition of the brain in the stroke protocol?

Materials and methods: you say "readers were blinded to the patients' clinical data", but then you say that the readers evaluated "any other CT findings which could explain the clinical presentation"...

Results: it might have been interesting to independently evaluate CT images and to assess concordance...

Discussion: ok

Bibliography: ok

Figures: an "a)" is missing in figure 2 caption. Figure 3a is wrong.  

Author Response

Reviewer 1

The article is interesting and well written.

Abstract: ok

Introduction: can you explain the rationale of routinely including a 180s post contrast acquisition of the brain in the stroke protocol?

The 180-second post-contrast acquisition is crucial for evaluating stroke mimics and assessing venous drainage. This delay enhances the differentiation between true ischemic events and conditions that mimic stroke. It also provides clearer visualization of venous structures, including both deep and superficial venous drainage, which are important prognostic factors (1,2).

  1. Parthasarathy R, Kate M, Rempel JL, et al. Prognostic Evaluation Based on Cortical Vein Score Difference in Stroke. Stroke. American Heart Association; 2013;44(10):2748–2754. doi: 10.1161/STROKEAHA.113.001231.
  2. Singh N, Bala F, Kim BJ, et al. Time-resolved assessment of cortical venous drainage on multiphase CT angiography in patients with acute ischemic stroke. Neuroradiology. 2022;64(5):897–903. doi: 10.1007/s00234-021-02837-1.

Materials and methods: you say "readers were blinded to the patients' clinical data", but then you say that the readers evaluated "any other CT findings which could explain the clinical presentation"...

The sentence is edited and the last sentence is suppressed upon your remark.

Results: it might have been interesting to independently evaluate CT images and to assess concordance...

Thank you for your remark. Ideally, independent evaluation of CT images would assess concordance. However, in our academic emergency radiology unit, readings are primarily done collaboratively between residents and radiologists. This approach ensures both educational and clinical needs are met, with experienced radiologists guiding the diagnostic process and training residents. Therefore, we have maintained the routinely used methodology.

Discussion: ok

Bibliography: ok

Figures: an "a)" is missing in figure 2 caption. Figure 3a is wrong.  

The Figure 3a has been modified upon your remark and figure 2 a has been renamed.

Reviewer 2 Report

Comments and Suggestions for Authors

After considering the manuscript, I found that the paper should undergo minor revisions and my comments are as follows.

1)      There is no literature review in the introduction. The introduction should be extended and also focused on the novelty of the study.

2)      The authors should identify the place of study and name of the hospital and the time of examinations.

3)      In the materials section, also some corrections about the protocol of CT examination are required. For example in lines 66-77 instead of a slash please use a dash between the numbers (for example 0.75/0.5 mm should be converted to 0.75-0.5 mm).

4)      The results should be more discussed.

5)      In Figure 1, the authors should provide a self-explanatory figure legend without the need to refer to the text.

6)      In the discussion, there is no comparison with the literature relevant to the work.

7)      Authors should prepare a conclusion section.

8)      The English needs some edits and corrections in the whole of the manuscript.

Comments on the Quality of English Language

The English needs some edits and corrections in the whole of the manuscript.

Author Response

After considering the manuscript, I found that the paper should undergo minor revisions and my comments are as follows.

1)      There is no literature review in the introduction. The introduction should be extended and also focused on the novelty of the study.

Edited upon your remark.

2)      The authors should identify the place of study and name of the hospital and the time of examinations.

This is edited upon your remark.

3)      In the materials section, also some corrections about the protocol of CT examination are required. For example in lines 66-77 instead of a slash please use a dash between the numbers (for example 0.75/0.5 mm should be converted to 0.75-0.5 mm).

This edited upon your remark.

4)      The results should be more discussed.

The text has been edited upon your remark

“Our study sought to evaluate whether VNCCT could replace NCCT for assessing patients admitted with suspected acute stroke. In this patient population, our results revealed an excellent correlation (intraclass correlation coefficients of 0.97 for the anterior circulation and 0.77 for the posterior circulation) for determining ASPECTS between VNCCT and NCCT [17]. This strong agreement suggests that both techniques are equivalent for this task. The slightly lower correlation coefficient in the posterior circulation can be attributed to the presence of multiple artifacts caused by the area's anatomy, which renders its analysis more challenging.

Additionally, we evaluated the diagnostic performance of both VNCCT and NCCT in detecting ischemia. VNCCT demonstrated a sensitivity of 41% (19 out of 46 patients) compared to 33% (15 out of 46 patients) for NCCT, indicating a superior ability to detect ischemic lesions. Specificity was similar between the two techniques, with VNCCT showing a specificity of 94% (97 out of 103 patients) and NCCT showing a specificity of 98% (101 out of 103 patients). These findings highlight the potential of VNCCT to enhance early ischemia detection, which is critical for timely and effective stroke management.

Moreover, our study identified six cases of intracerebral hemorrhage, all detected by both VNCCT and NCCT, confirming that VNCCT is also reliable for identifying hemorrhagic events. The capability of VNCCT to provide equivalent diagnostic information without the need for separate NCCT scans suggests a more efficient workflow, reducing the need for multiple scans and thereby minimizing patient movement and scan time.”

5)      In Figure 1, the authors should provide a self-explanatory figure legend without the need to refer to the text.

Figure 1 legend is added upon your remark

Figure 1: This flowchart illustrates the diagnostic outcomes of the study.

- Acute Ischemia: Found in 46 patients (30.5%).

  -Anterior Circulation: Found in 39 patients (85.5%); detected in 16 patients on virtual non-contrast CT (VNCCT) and 12 on non-contrast CT (NCCT).

  - Posterior Circulation: Found in 8 patients (15.5%); detected in 6 patients on VNCCT and 6 on NCCT. Note: One patient exhibited both anterior and posterior ischemic signs.

- Intraparenchymal Hemorrhage: Found in 6 patients (4%), all detected on both VNCCT and NCCT.

- Alternative Diagnoses: Found in 11 patients (11%); 5 detected on VNCCT and 5 on NCCT.

- Normal Findings: Present in 85 patients (89%), confirmed by both VNCCT and NCCT.

6)      In the discussion, there is no comparison with the literature relevant to the work.

Discussion has been edited to integrate literature relevant to the work.

“These findings align with previous studies on dual-energy CT (DECT) in ischemic stroke. For instance, a study comparing DECT with conventional NCCT reported a concordance correlation coefficient (CCC) of 0.44 for VNCCT, higher than the 0.23 for conventional NCCT, highlighting VNCCT's superior sensitivity in detecting early ischemic changes. Another study found that DECT could effectively differentiate between iodine and blood components, enhancing the detection of ischemic tissue and providing a reliable alternative to NCCT​ [23].

Additionally, our results are consistent with the broader literature on DECT's prognostic value. A meta-analysis indicated that patients with contrast staining (CS) detected by DECT had a higher risk of hemorrhagic transformation (HT) and poorer functional outcomes at 90-day follow-up compared to those without CS. This underscores DECT's utility in identifying patients at higher risk of complications, thus aiding in more informed clinical decision-making​ [24].”

7)      Authors should prepare a conclusion section.

Conclusion section has been prepared and added upon your remark.

“Conclusion

Our study suggests that VNCCT could safely replace NCCT for assessing patients with suspected acute stroke. The high intraclass correlation coefficients for ASPECTS in both the anterior and posterior circulation indicate strong agreement between VNCCT and NCCT. Additionally, VNCCT demonstrated better sensitivity and comparable specificity to NCCT in detecting acute ischemia. These findings highlight the potential of VNCCT to streamline the diagnostic process, reduce patient movement, and decrease scan time without compromising diagnostic accuracy. Further studies with larger sample sizes are warranted to confirm our preliminary positive observations and to explore the practical implementation of VNCCT in various clinical settings.”

8)      The English needs some edits and corrections in the whole of the manuscript.

The manuscript has been corrected and edited by a native English speaker : Prof Remy Guillevin